**Data Availability Statement:** All relevant data are within the manuscript and its Supporting information files.

# Monitoring adherence to pharmacological therapy and follow-up examinations among patients with type 2 diabetes in community pharmacies. Results from an experience in Italy

**Teresa Spadea**[1], **Roberta Onorati**[1], **Francesca Baratta**[2]*, **Irene Pignata**[2], **Marco Parente**[3], **Lavinia Pannacci**[4], **Domenica Ancona**[5], **Paola Ribecco**[6], **Giuseppe Costa**[1,7], **Roberto Gnavi**[1], **Paola Brusa**[2]

1 Epidemiology Unit, ASL TO3, Piedmont Region, Grugliasco, Italy, 2 Department of Drug Science and Technology, University of Turin, Turin, Italy, 3 Federfarma Torino, Turin, Italy, 4 Prevention Unit, Umbria Region, Perugia, Italy, 5 Pharmaceutical Department ASL BAT, Puglia Region, Trani, Italy, 6 Federfarma Brindisi, Brindisi, Italy, 7 Department of Clinical and Biological Sciences, University of Turin, Turin, Italy

* francesca.baratta@unito.it

## Abstract

### Introduction

Type 2 diabetes is an important public health issue, yet adherence to drugs and regular clinical follow-up is still suboptimal. This study aims to evaluate a community pharmacy programme for monitoring and enhancing adherence to prescribed pharmacological therapies and recommended examinations among patients with confirmed diabetes.

### Methods

The intervention was conducted in different Italian areas between April 2017 and January 2018. All adult patients who entered a pharmacy with a personal prescription for any antidiabetic drug and agreed to participate, were interviewed. Those found to be non-adherent received counselling from the pharmacists. All patients were invited for a follow-up interview after 3 months.

### Results

Overall, 930 patients were enrolled and completed the baseline interview. We found low rates of non-adherence, ranging from 8% to 13% for prescribed pharmacological therapies, and 11–29% for the recommended clinical examinations. Non-adherence to oral therapies was higher among younger and recently diagnosed patients; that to clinical examinations was higher in men, those with an intermediate duration of diabetes and less educated patients. Large geographical differences persisted after the adjustment for individual factors. Only 306 patients (32.9%) returned for the follow-up interview, most of whom were already adherent at baseline.

**Funding:** This project, "La farmacia dei servizi per il controllo delle patologie croniche: sperimentazione e trasferimento di un modello di intervento di prevenzione sul diabete tipo 2", has been realized with the financial support of the Italian Ministry of Health – CCM (National Center for Disease Prevention and Control) 2015.

**Competing interests:** The authors have declared that no competing interests exist.

## Conclusions

Poor adherence to drugs or clinical examinations is not easy to identify in the usual operating setting of community pharmacies. Furthermore, the majority of patients did not return for follow-up, making it impossible to evaluate the efficacy of the pharmacists' counselling. It might be more effective to plan interventions addressed to specific subgroups of patients or areas.

## Introduction

Both the number of people with diabetes and its prevalence are dramatically increasing worldwide. About 3.4 million Italians are affected by diabetes and, overall, its prevalence is 5%. However, its distribution is uneven; it is higher in the south of the country than in the north, higher in poorly educated people than in the highly educated, and higher in men than in women [1]. Due to its burden in terms of social and health costs, the disease represents an important public health issue. Despite the increased awareness of diabetes and its complications, a non-negligible number of patients are still undertreated or do not adhere to clinical guidelines [2,3]. The available literature reports that adherence to drugs ranges from 20% to 80% [4] while adherence to the glycated haemoglobin and cholesterol tests rarely exceeds 70% [3,5–8].

Adherence and persistence to therapies, as well as compliance to regular monitoring and clinical follow-up are the main tertiary prevention actions associated to better outcomes, a reduced or delayed onset of complications, and, not least, to reduced expenditure [2,9–13]. As a consequence, it is of paramount importance that effective strategies to find non-adherent patients and improve their compliance to guidelines are identified. Community pharmacies may be one of the settings where these actions can be carried out, as has already been reported in other studies [14–16].

In Italy, there are more than 19,000 community pharmacies. As nearly every municipality has at least one pharmacy, which are easily accessible and free of charge, they are used by the population as a fast and trustworthy gateway to health services, and as a contact point with the health care system [17,18]. In 2012, the Piedmont Regional Orders of Pharmacists, Federfarma Piemonte and the University of Turin launched an extensive programme aimed to counteract the negative effects of non-communicable diseases [18–23]. The programme for diabetes was based upon two main preventive actions: 1. the identification of undiagnosed cases of the disease among customers of community pharmacies (secondary prevention); and, 2. monitoring and enhancing adherence to pharmacological treatment and follow-up guidelines among people with confirmed diabetes (tertiary prevention). We have already reported the general results of the regional programme [18], and specifically those of the impact of the opportunistic screening [23]. Subsequently, in 2015, the Italian Health Ministry funded a study to assess the transferability and efficacy of the programme in the setting of community pharmacies in other regional contexts.

In this paper, we report the results of the second action of the preventive programme, with the twofold objective of monitoring adherence to prescribed drug therapies and to the examinations recommended by clinical guidelines, and assessing the impact of the intervention. We also discuss the implications of these results in terms of public health.

## Materials and methods

### Study population and intervention protocol

The intervention was conducted in two regions in Central and Southern Italy (Umbria and Puglia, respectively). The study involved a territory comprised of three health districts in Umbria and two provinces in Puglia (Barletta-Andria-Trani (BAT) and Brindisi), that in total

care for about 1 million inhabitants. The study consisted of two steps: the first was a cross-sectional survey aimed at identifying patients with diabetes that were non-adherent to either their prescribed therapies or their regular clinical examinations, and investigating their characteristics; the second step was designed as a follow-up study of all the interviewees (both adherent and non-adherent subjects) to assess the impact of the intervention.

All of the pharmacists operating in private and public community pharmacies in the territories were invited to participate in the project on a voluntary basis. Those who agreed were enrolled in a training course on diabetes (conducted by a senior diabetologist) and on the study procedures and instruments, with special attention being paid to the questionnaires, to ensure that all pharmacists collected data homogeneously.

Over the period April 2017-January 2018 all adult persons who entered a pharmacy with a personal prescription for any antidiabetic drug were informed of the aims of the study and invited to participate. Given the expected low number of daily entries of diabetic patients, no sampling was applied. Those who agreed gave their written informed consent to be interviewed and followed-up. Individuals that reported that they suffered from type 1 diabetes or that they were to have their first prescription were excluded. The pharmacists interviewed the participants in a consultation room within each pharmacy and then invited them to repeat the interview in the same pharmacy after 3 months, to assess any change.

Adherence to prescribed drug therapies was investigated using a 4-item scale, developed from the Italian version of the original 8-item Morisky scale [24]. The questionnaire also enquired as to whether the patient had had access to any emergency room or hospitalization; asked if he suffered from any comorbidities such as dyslipidaemia, hypertension and heart failure; and investigated adherence to all classes of medications taken by the patient. A second questionnaire explored adherence and the correct timing of eight clinical examinations, recommended by the Italian Association of Diabetologists and the Italian Society of Diabetology [25]. Finally, we collected information on education, social/family support and household composition. Educational level, measured as the maximum attained qualification, was categorized in three classes: low, including no formal education and primary school (corresponding to the UNESCO International Standard Classification of Education 1997 (ISCED97) levels 0–1); medium, i.e. middle and vocational school (ISCED97 levels 2-3C); and high, including high school and university degrees (ISCED97 levels 3A, 3B, 5 and 6) [26]. Household condition and social support were represented by two dichotomous variables indicating whether the patient lived alone and could receive help in case of need. The questionnaires are reported in the S1 File. All data were collected electronically and stored in a central database.

All individuals who resulted non-adherent (for either drugs or visits) received counselling on correct medicine taking and the timing for recommended examinations. Moreover, as agreed with the local representatives of general medicine trade unions and professional orders, patients who declared that they had not carried out all follow-up checks were referred to their general practitioner (GP) for possible further checks.

At the end of the project, a short online satisfaction questionnaire was administered to the pharmacists in order to collect their qualitative evaluations on the effectiveness of the intervention. They were asked to indicate three positive aspects and three negative aspects of the project. Furthermore, they could add suggestions for future studies. All the items were free-text open questions (S2 File).

## Outcome definition and recording

A patient with a score of the 4-item scale higher than one was classified as non-adherent to the prescribed pharmacological therapies; patients using both insulin and oral drugs were

included in both therapy groups. In the case of clinical examinations, we first considered non-adherence separately for each item explored. Given that the percentage of patients who responded positively to all questions was very low (19%) and considering that the recommendations do not have the same clinical weight, we decided to analyse in detail only the measurement of glycated haemoglobin (HbA1c) every 6 months, as this is the main indicator of disease control. Furthermore, we calculated the Guideline Composite Indicator (GCI), which is a comprehensive indicator of adherence that has proven to correlate with more favourable health outcomes [11], and which classifies patients who have not carried out the HbA1c test and at least two checks from cholesterol, albuminuria and fundus of the eye, as being non-adherent.

## Statistical analysis

We used the Chi squared tests for categorical variables to assess differences across study areas; a 2-tail p-value of less than 0.05 was considered statistically significant. Determinants of non-adherence were investigated performing robust Poisson multivariable regression models, which estimate prevalence ratios (PR) with their 95% confidence intervals (95% CI) [27]. All statistical analyses were run using SAS-ver.9.3 and STATA-ver.10.

## Ethics statement

The study has been approved by the Italian Ministry of Health as part of the CCM (National Center for Disease Prevention and Control) programme 2015, and, according to Italian legislation, does not require further evaluation by the Ethics Committee. Nonetheless, the same protocol had been approved by the "Azienda Sanitaria Locale ASLTO2" Ethics Committee, Approval Protocol n°46480/2013 [23].

## Results

Overall, at least one pharmacist from 155 out of the 253 (61.3%) pharmacies in the study areas attended the training course with at least one pharmacist (248 pharmacists completed the training). Of these, 99 (64% of trained pharmacies) participated in the programme, enrolling 1037 patients. Of these patients, 62 (6%) were affected by Type 1 diabetes and were excluded, while 45 (4.3%) refused to participate (the complete flow chart is reported in the S3 File). Table 1 shows the characteristics of the remaining 930 patients, overall and by centre. Men were 59% of the population, two thirds of participants were over 64 years of age, and about 60% had a duration of diabetes longer than 5 years; 85% had at least one comorbidity. As for the sociodemographic indicators, 43% were low educated patients, while 20% had at least a high school diploma. The great majority (about 90%) did not live alone and could receive help in case of need.

The prevalence of non-adherence to pharmacological therapies and to clinical examinations, according to individual characteristics, are reported in Table 2. Looking at therapies, out of 261 insulin users (28% of patients entering the pharmacies), only 20 (7.7%) were non-adherent, while among the 836 patients with a prescribed oral therapy (90% of patients) the prevalence of non-adherence raised to 12.7%. When comparing insulin and oral drug users, the distribution of non-adherence resulted reversed for most individual characteristics, making a pooled analysis impossible. Therefore, given the small number of non-adherent insulin users, we only focussed on non-adherence to oral drugs in the subsequent analyses; prevalence was higher among women, younger patients, those with a lower duration of disease, without comorbidities, and in socially advantaged patients (more educated, who had social support and who did not live alone).

**Table 1. Characteristics of the enrolled patients with type 2 diabetes, by study area and overall.**

| | Umbria Region (n = 330) | | BAT Province (n = 380) | | Brindisi Province (n = 220) | | Chi-square | ALL (n = 930) | |
|---|---|---|---|---|---|---|---|---|---|
| | n | % | n | % | n | % | p-value* | n | % |
| **Gender** | | | | | | | | | |
| *Women* | 126 | 38.2 | 156 | 41.1 | 96 | 43.6 | 0.433 | 378 | 40.6 |
| *Men* | 204 | 61.8 | 224 | 58.9 | 124 | 56.4 | | 552 | 59.4 |
| **Age** | | | | | | | | | |
| *<45* | 0 | 0 | 13 | 3.4 | 7 | 3.2 | 0.002 | 20 | 2.2 |
| *45–54* | 22 | 6.7 | 38 | 10 | 29 | 13.2 | | 89 | 9.6 |
| *55–64* | 71 | 21.5 | 89 | 23.5 | 50 | 22.7 | | 210 | 22.6 |
| *≥65* | 237 | 71.8 | 239 | 63.1 | 134 | 60.9 | | 610 | 65.7 |
| **Duration of diabetes** (years) | | | | | | | | | |
| *<1* | 22 | 6.7 | 29 | 7.6 | 16 | 7.3 | 0.012 | 67 | 7.2 |
| *1–5* | 81 | 24.6 | 88 | 23.2 | 56 | 25.4 | | 225 | 24.2 |
| *5–10* | 81 | 31.2 | 102 | 26.8 | 58 | 26.4 | | 241 | 25.9 |
| *>10* | 103 | 24.5 | 142 | 37.4 | 78 | 35.5 | | 323 | 34.7 |
| *Unknown* | 43 | 13 | 19 | 5 | 12 | 5.4 | | 74 | 8 |
| **Comorbidities** | | | | | | | | | |
| *Yes* | 281 | 85.1 | 319 | 83.9 | 194 | 88.2 | 0.363 | 794 | 14.6 |
| *No* | 49 | 14.9 | 61 | 16.1 | 26 | 11.8 | | 136 | 85.4 |
| **Educational level** | | | | | | | | | |
| *High* | 75 | 22.7 | 73 | 19.2 | 38 | 17.3 | 0.167 | 186 | 20 |
| *Medium* | 126 | 38.2 | 129 | 34 | 89 | 40.5 | | 344 | 36.9 |
| *Low* | 129 | 39.1 | 178 | 46.8 | 93 | 42.3 | | 400 | 43.1 |
| **Living alone**** | | | | | | | | | |
| *Yes* | 35 | 10.6 | 42 | 11.1 | 16 | 7.3 | 0.296 | 93 | 10.1 |
| *No* | 294 | 89.4 | 338 | 88.9 | 204 | 92.7 | | 836 | 89.9 |
| **Social network** | | | | | | | | | |
| *Yes* | 312 | 94.6 | 351 | 92.4 | 192 | 87.3 | 0.008 | 855 | 91.9 |
| *No* | 18 | 5.4 | 29 | 7.6 | 28 | 12.7 | | 75 | 8.1 |

* p-value <0.05 indicates a statistical significant difference between areas.

** 1 case missing.

Looking at the recommended clinical examinations, around 24% of patients were not adherent to HbA1c, and similar percentages were observed for albuminuria and the eye examination; only 11% did not check their cholesterol level. When the comprehensive GCI was considered, non-adherence increased to 29%. Non-adherence is generally greater in men, in those without comorbidities and in the less educated patients. With respect to the duration of diabetes, the prevalence has a reversed U-shaped curve, with lowest levels in the newly diagnosed cases and in those with long durations. Finally, we observed very large geographical differences in all the indicators of non-adherence, with lower rates in the central area of the country (Umbria) and the highest in the Brindisi Province.

After the multivariable adjustment (Table 3), non-adherence to oral therapies remained associated only with age and duration of diabetes, increasing with decreasing age and decreasing duration of the disease. No significant differences emerged with regards to gender or social indicators. Non-adherence to clinical examinations was significantly higher in men and in patients with intermediate duration of diabetes. However, a higher risk of non-adherence was

**Table 2. Number of cases and prevalence of non-adherence to prescribed pharmacological therapies and recommended clinical examinations by patient characteristics.**

| | Pharmacological therapies | | | | Clinical examinations | | | | | | | | | | |
| --- | --- | --- | --- | --- | --- | --- | --- | --- | --- | --- | --- | --- | --- | --- | --- |
| | Insulin | | Oral therapy | | HbA1c | | Cholesterol | | Albuminuria | | Fundus eye | | GCI | |
| | (n = 261) | | (n = 836) | | (n = 930) | | (n = 930) | | (n = 930) | | (n = 930) | | (n = 930) | |
| | n | % | n | % | n | % | n | % | n | % | n | % | n | % |
| TOTAL | 20 | 7.7 | 106 | 12.7 | 222 | 23.9 | 106 | 11.4 | 216 | 23.2 | 224 | 24.1 | 272 | 29.2 |
| **Gender** | | | | | | | | | | | | | | |
| Women | 6 | 6.2 | 48 | 13.8 | 83 | 22.0 | 43 | 11.4 | 90 | 23.8 | 92 | 24.3 | 102 | 27.0 |
| Men | 14 | 8.5 | 58 | 11.9 | 139 | 25.2 | 63 | 11.4 | 126 | 22.8 | 132 | 23.9 | 170 | 30.8 |
| **Age** | | | | | | | | | | | | | | |
| <45 | 0 | 0.0 | 6 | 33.3 | 5 | 25.0 | 3 | 15.0 | 5 | 25.0 | 8 | 40.0 | 6 | 30.0 |
| 45–54 | 1 | 3.4 | 20 | 26.3 | 17 | 19.1 | 8 | 9.0 | 21 | 23.6 | 29 | 32.6 | 27 | 30.3 |
| 55–64 | 4 | 7.1 | 26 | 13.1 | 47 | 22.4 | 20 | 9.5 | 39 | 18.6 | 46 | 21.9 | 54 | 25.7 |
| ≥65 | 15 | 8.7 | 54 | 9.9 | 152 | 24.9 | 74 | 12.1 | 150 | 24.6 | 140 | 23.0 | 184 | 30.2 |
| **Duration of diabetes** (years) | | | | | | | | | | | | | | |
| <1 | 0 | 0.0 | 12 | 18.8 | 13 | 19.4 | 9 | 13.4 | 20 | 29.9 | 29 | 43.3 | 20 | 29.9 |
| 1–5 | 3 | 8.3 | 42 | 19.8 | 64 | 28.4 | 25 | 11.1 | 59 | 26.2 | 68 | 30.2 | 78 | 34.7 |
| 5–10 | 5 | 8.3 | 18 | 8.0 | 63 | 26.1 | 34 | 14.1 | 62 | 25.7 | 57 | 23.7 | 77 | 32.0 |
| >10 | 12 | 8.3 | 27 | 10.3 | 66 | 20.4 | 33 | 10.2 | 55 | 17.0 | 48 | 14.9 | 73 | 22.6 |
| Unknown | 0 | 0.0 | 6 | 15.4 | 16 | 21.6 | 5 | 6.8 | 20 | 27.0 | 22 | 29.7 | 24 | 32.4 |
| **Comorbidities** | | | | | | | | | | | | | | |
| Yes | 19 | 8.0 | 93 | 12.5 | 186 | 23.4 | 83 | 10.5 | 177 | 22.3 | 181 | 22.8 | 225 | 28.3 |
| No | 1 | 4.2 | 13 | 14.3 | 36 | 26.5 | 23 | 16.9 | 39 | 28.7 | 43 | 31.6 | 47 | 34.6 |
| **Educational level** | | | | | | | | | | | | | | |
| High | 7 | 12.1 | 26 | 15.9 | 37 | 19.9 | 18 | 9.7 | 36 | 19.4 | 34 | 18.3 | 46 | 24.7 |
| Medium | 3 | 3.6 | 37 | 11.7 | 73 | 21.2 | 35 | 10.2 | 71 | 20.6 | 83 | 24.1 | 92 | 26.7 |
| Low | 10 | 8.4 | 43 | 12.1 | 112 | 28.0 | 53 | 13.3 | 109 | 27.3 | 107 | 26.8 | 134 | 33.5 |
| **Living alone** | | | | | | | | | | | | | | |
| Yes | 3 | 10.7 | 9 | 10.7 | 23 | 24.7 | 13 | 14.0 | 22 | 23.7 | 20 | 21.5 | 26 | 28.0 |
| No | 16 | 6.9 | 97 | 12.9 | 199 | 23.8 | 93 | 11.1 | 194 | 23.2 | 204 | 24.4 | 246 | 29.4 |
| **Social network** | | | | | | | | | | | | | | |
| Yes | 20 | 8.3 | 101 | 13.2 | 209 | 24.4 | 95 | 11.1 | 199 | 23.3 | 199 | 23.3 | 253 | 29.6 |
| No | 0 | 0.0 | 5 | 6.9 | 13 | 17.3 | 11 | 14.7 | 17 | 22.7 | 25 | 33.3 | 19 | 25.3 |
| **Study area** | | | | | | | | | | | | | | |
| BAT | 5 | 5.0 | 46 | 13.2 | 100 | 26.3 | 45 | 11.8 | 99 | 26.1 | 103 | 27.1 | 124 | 32.6 |
| BR | 7 | 10.6 | 36 | 18.4 | 74 | 33.6 | 32 | 14.6 | 72 | 32.7 | 69 | 31.4 | 87 | 39.6 |
| Umbria | 8 | 8.4 | 24 | 8.2 | 48 | 14.6 | 29 | 8.8 | 45 | 13.6 | 52 | 15.7 | 61 | 18.5 |

observed among patients who did not report the duration of their disease. Non-adherence was also higher in the least educated patients. Large geographical differences persisted after the adjustment for all the other individual factors.

After three months, only 250 patients (26.9% of those invited) returned for a follow-up interview; if we include the 56 people who returned beyond the time limit, the percentage increased to 32.9%. Most of the returnees were already adherent at baseline (97% of insulin users and 91% of oral drug users),yet adherence showed a small increase at follow-up, remaining at 97% among insulin users and reaching 94% among oral drug users. Unfortunately, only 2 non-adherent insulin users and 14 patients non-adherent to oral drugs returned for follow-

**Table 3. Individual characteristics associated to non-adherence to prescribed oral therapy and recommended clinical examinations.**

| | Oral therapy (n = 834) | | HbA1c (n = 930) | | GCI (n = 930) | |
| --- | --- | --- | --- | --- | --- | --- |
| | PR | 95% CI | PR | 95% CI | PR | 95% CI |
| **Gender** | | | | | | |
| *Women* | 1 | | 1 | | 1 | |
| *Men* | 0.86 | 0.59–1.27 | 1.27 | 0.99–1.63 | **1.25** | **1.01–1.56** |
| **Age** | | | | | | |
| *≥65* | 1 | | 1 | | 1 | |
| *55–64* | 1.25 | 0.79–1.97 | 0.92 | 0.69–1.23 | 0.86 | 0.66–1.11 |
| *45–54* | **2.23** | **1.36–3.65** | 0.74 | 0.47–1.16 | 0.95 | 0.68–1.33 |
| *<45* | 2.14 | 0.99–4.64 | 0.85 | 0.40–1.83 | 0.84 | 0.42–1.65 |
| **Duration of diabetes** (years) | | | | | | |
| *>10* | 1 | | 1 | | 1 | |
| *6–10* | 0.77 | 0.44–1.35 | 1.33 | 0.99–1.80 | **1.46** | **1.12–1.91** |
| *1–5* | **1.74** | **1.10–2.76** | **1.55** | **1.15–2.08** | **1.67** | **1.28–2.17** |
| *<1* | 1.54 | 0.82–2.92 | 0.94 | 0.55–1.63 | 1.34 | 0.87–2.05 |
| *Unknown* | 1.08 | 0.48–2.43 | 1.28 | 0.80–2.03 | **1.72** | **1.20–2.49** |
| **Comorbidities** | | | | | | |
| *No* | 1 | | 1 | | 1 | |
| *Yes* | 1.03 | 0.62–1.71 | 0.77 | 0.56–1.07 | 0.81 | 0.61–1.08 |
| **Educational level** | | | | | | |
| *High* | 1 | | 1 | | 1 | |
| *Medium* | 0.72 | 0.45–1.14 | 1.10 | 0.77–1.55 | 1.10 | 0.82–1.49 |
| *Low* | 0.92 | 0.55–1.54 | **1.49** | **1.06–2.10** | **1.46** | **1.09–1.96** |
| **Living alone** | | | | | | |
| *No* | 1 | | 1 | | 1 | |
| *Yes* | 0.85 | 0.44–1.64 | 1.09 | 0.75–1.60 | 1.01 | 0.71–1.42 |
| **Social network** | | | | | | |
| *Yes* | 1 | | 1 | | 1 | |
| *No* | 0.47 | 0.20–1.14 | 0.62 | 0.38–1.03 | 0.77 | 0.53–1.13 |
| **Study area** | | | | | | |
| *BAT* | 1 | | 1 | | 1 | |
| *BR* | 1.31 | 0.88–1.95 | **1.35** | **1.06–1.75** | **1.25** | **1.00–1.56** |
| *Umbria* | 0.66 | 0.40–1.07 | **0.55** | **0.40–0.75** | **0.55** | **0.42–0.72** |

Prevalence Ratios (PR) estimated by multivariate robust Poisson models.

up. Therefore, the planned evaluation of the effectiveness of counselling for non-adherent patients cannot be performed.

As a further qualitative evaluation, we analysed the satisfaction questionnaires completed at the end of the study by almost 70 pharmacists, out of the 99 participating in the programme. The pharmacists reported general appreciation for having been involved in the project, confirming that they had acquired new instruments in pharmaceutical care and had improved their relationship with their customers. They also reported that patients appreciated this free and customized intervention that made them feel cared for. The most frequently raised criticism was the difficulty in involving GPs.

## Discussion

### Summary of results

This study confirms that community pharmacies and pharmacists are a good setting for conducting investigations as their project participation is generally very high, as has already been reported [23,28]. Moreover, more than nine hundred diabetic patients were intercepted and agreed to be interviewed.

In the populations covered by the study, we found very low rates of non-adherence to prescribed pharmacological treatments, ranging from an average of 8% for insulin to 13% for oral antidiabetic drugs. Conversely, rates of adherence to the clinical examinations recommended by the guidelines for follow-up–although they were on average above 70%–may be improved, particularly with regards to the combination of different examinations, as captured by the composite indicator (non-adherence goes from 11 to 29%). Age and duration of diabetes were the most significant predictors of non-adherence, but also educational level and geographical area had an independent impact.

### Possible explanations

Previous population-based studies had reported adherence to drugs ranging from 20% to 80% [4], and from 28% to 36% when the composite indicator of clinical follow-up was considered [3,5,6]. Higher adherence rates have been reported for the HbA1c and cholesterol tests, taken singularly, but they hardly reached such high values as in our study [3,5–8]. Compared to adherence estimated at the population level, our data show that patients who go to a pharmacy are likely to be selected among those who are more adherent and therefore less in need of a reinforcement intervention, particularly for drug therapies. Although this was a foreseen intrinsic characteristic of the enrolment strategy (patients entering a pharmacy to acquire drugs for diabetes), previous experiences in Piedmont had shown a higher prevalence of non-adherence [18,29], leaving more space for improvement via the professional counselling of pharmacists. Unfortunately, most of the non-adherent patients at baseline did not return to the pharmacy for the requested feedback, therefore it was not possible to measure and evaluate the effect of the counselling.

As regards the impact of age and duration of disease on adherence, the literature is not consistent, mainly because of the multifaceted nature of the phenomenon; therefore, disentangling the impact of single factors is difficult [30,31]. In our study, non-adherence to prescribed oral therapies is higher in young and recently diagnosed subjects. Awareness of one's health may actually increase with age and disease duration, but a selective survival mechanism cannot be excluded; less adherent patients may have died earlier and are therefore not found among the older patients or with longer durations of disease. Our results also suggest the existence of an intermediate period of disease duration, during which patients have a decline in their attention to controlling their disease, which was also shown for the recommended clinical examinations. It might therefore be more effective to plan possible reinforcement interventions in relation to the duration of the disease, in order to underly the importance of maintaining high adherence to both drugs and clinical follow-up since the first diagnosis. Analogously, more counselling could be specifically addressed to younger patients.

We observed large geographical differences, with higher levels of adherence in Umbria, which suggest that either much more selective recruitment or higher adherence levels were present in the catchment area of the participating pharmacies. Indeed, the different organization of diabetic patient care in the different local health systems may also explain the geographical differences, particularly in clinical follow-up. Whatever mechanism is in place, high

adherence rates provide little room for improvement, meaning that any such intervention would have low efficiency. This therefore suggests the need to identify in advance the areas where such an intervention would result to be more effective.

Interestingly, we observed a significant excess of non-compliance to the recommended clinical follow-up examinations among people with lower educational qualifications. This indicates that interventions aimed at increasing adherence to a correct therapeutic pathway could be specifically tailored towards less educated patients, and thus may contribute to reducing inequalities in the negative outcomes of the disease.

A further interesting result that should be considered in the overall evaluation of the programme is that pharmacists showed great satisfaction and felt that they had improved their relationship with customers with the benefit of their greater loyalty to the pharmacy.

## Limitations and strengths

The main limitation of this study, as has already been discussed, was the enrolment of patients with high levels of adherence and, moreover, the difficulty faced in tracing and making patients return to the pharmacy for follow-up, particularly in the case of the few non-adherent patients at baseline. This suggests that monitoring and enhancing adherence do not work properly in the usual operating setting of the pharmacy. In similar future projects, it would be necessary to enrol patients using strategies that identify subjects at higher risk of non-adherence or to shift focus by addressing patients at risk of therapeutic inappropriateness. This could be achieved via improved interaction between GPs and pharmacies in order to build integrated care pathways that include all the actors of primary care. One strength of this project was, indeed, the use of common software for collecting data in a harmonized database in all pharmacies; the same platform could be used for sharing patient information between health professionals.

A possible bias may derive from the instrument used to assess drug adherence. Indeed, similarly to what suggested in the literature [32], we used a 4-item questionnaire because of its brevity and acceptability; on the other hand, it investigates only some macro-aspects of non-adherence, such as having forgotten or voluntarily interrupted taking drugs, and this may have underestimated the number of non-adherent patients. However, in a subsequent sensitivity analysis, we found that even with a lower non-adherence cut-off (score> 0, hence higher rates), multivariable models on the determinants of non-adherence yielded substantially the same results. The original 8-item Morisky questionnaire [33], which also enquires as to specific situations of occasional non-adherence, or other possible tools could be tested in future studies.

## Conclusions

This study aimed to assess the transferability and efficacy of a community pharmacy programme for tertiary prevention among patients with type 2 diabetes. It provided us with some key information, which could be useful for future intervention planning.

The first lesson learned from our study is that the enrolment of subjects with poor adherence to drug therapies is not easy in the usual operating setting of pharmacies. Therefore, it is necessary to improve the methods for identifying therapeutic inappropriateness and intercepting non-adherent patients. On the other hand, while non-adherence to recommended clinical examinations is more easily identifiable in community pharmacies, it is necessary to strengthen collaboration with GPs if corrective mechanisms are to be found. Analogously, the involvement of different areas of the country has allowed us to understand the importance of planning similar interventions in areas where possible problems of adherence to therapeutic

pathways are highlighted in advance, in order to maximize the preventive impact of pharmacists' counselling.

Secondly, the planned evaluation of the effectiveness of counselling for non-adherent patients could not be fully performed. Indeed, only one third of patients returned to the pharmacies for follow-up. Furthermore, most of these were already adherent at baseline, making it impossible to evaluate any possible change in adherence following the pharmacist's intervention. Nonetheless, pharmacists reported an improved relationship with their customers, which suggests that similar programmes, developed within community pharmacies, could act as a lever to improve patient confidence and loyalty, and ensure the greater effectiveness of pharmacists' counselling.

Finally, we should recall that these are the results of a "stand-alone" intervention of pharmacists. The full potential of the involvement of pharmacists in the health service could be exploited in structured territorial processes of support to chronic patients.

## Supporting information

**S1 File. Patient questionnaires.**
(DOC)

**S2 File. Pharmacists satisfaction questionnaire.**
(DOCX)

**S3 File. Study flow chart.**
(PDF)

**S4 File. Excel file with data.**
(XLSX)

## Acknowledgments

We thank all the other colleagues who participated in this project: Massimo Mana (Federfarma Piemonte), Paolo Cavallo Perin (University of Turin), Mariangela Rossi and Gianni Giovannini (Umbria Region), Silvia Pagliacci and Valentina Furbini (Federfarma Umbria), Crescenzo La Forgia (ASL BAT), Michele Pellegrini Calace and Stefano Vitti (Federfarma BAT), Salvatore Leo (Federfarma Brindisi). We are grateful to Luigi D'Ambrosio Lettieri (Order of Pharmacists of the Bari province) for supporting the project and ensuring hospitality in Puglia. Special thanks go to Ms. Rosaria Foggetti (Epidemiology ASL TO3) for her invaluable work in the administrative management of the project.

We also thank the Regional Orders of Pharmacists for their contribution to the implementation of the project and Ezio Festa (ATF Informatics, Cuneo, Italy) for developing the data-gathering software and database management.

Finally, we must acknowledge the precious work of all the pharmacists, without whom we could not have carried out this study.

## Author Contributions

**Conceptualization:** Teresa Spadea, Giuseppe Costa, Roberto Gnavi, Paola Brusa.

**Data curation:** Teresa Spadea, Roberta Onorati.

**Formal analysis:** Roberta Onorati.

**Funding acquisition:** Giuseppe Costa.

**Investigation:** Teresa Spadea, Roberta Onorati, Francesca Baratta, Irene Pignata, Marco Parente, Roberto Gnavi, Paola Brusa.

**Project administration:** Teresa Spadea, Paola Brusa.

**Supervision:** Lavinia Pannacci, Domenica Ancona, Paola Ribecco.

**Writing – original draft:** Teresa Spadea, Roberta Onorati, Francesca Baratta, Irene Pignata, Roberto Gnavi.

**Writing – review & editing:** Teresa Spadea, Francesca Baratta, Irene Pignata, Roberto Gnavi.

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
