## [Decision Letter · Decision Letter 0]

20 May 2021

PONE-D-21-09684

Monitoring adherence to clinical guidelines among patients with type 2 diabetes in community pharmacies. Results from an experience in Italy

PLOS ONE

Dear Dr. Baratta,

Thank you for submitting your manuscript to PLOS ONE. After careful consideration, we feel that it has merit but does not fully meet PLOS ONE’s publication criteria as it currently stands. Therefore, we invite you to submit a revised version of the manuscript that addresses the points raised during the review process.

We look forward to receiving your revised manuscript.

Kind regards,

Filipe Prazeres, MD, MSc, Ph.D.

Academic Editor

PLOS ONE

Journal Requirements:

2. We note that you used the Morisky Medication Adherence Scale (MMAS-8) in your study. It is our understanding that this scale is protected by copyright and requires a license agreement for use. Please explain in your Methods section whether you obtained permission and a license agreement for the use of the MMAS-8 in your study. In addition, we note that the scale is reproduced as Supporting Information. Due to copyright restrictions, please remove this from your submission.

Reviewers' comments:

Reviewer's Responses to Questions

**Comments to the Author**

1. Is the manuscript technically sound, and do the data support the conclusions?

Reviewer #1: Yes

Reviewer #2: Partly

Reviewer #3: Yes

2. Has the statistical analysis been performed appropriately and rigorously? 

Reviewer #1: Yes

Reviewer #2: No

Reviewer #3: Yes

3. Have the authors made all data underlying the findings in their manuscript fully available?

Reviewer #1: Yes

Reviewer #2: No

Reviewer #3: Yes

4. Is the manuscript presented in an intelligible fashion and written in standard English?

Reviewer #1: Yes

Reviewer #2: No

Reviewer #3: Yes

5. Review Comments to the Author

Reviewer #1: Dear Authors,

Manuscript, named as “Monitoring adherence to clinical guidelines among patients with type 2 diabetes in community pharmacies. Results from an experience in Italy” was sended me to review.

Study is very valuable in terms of study design and contribution to the literature.

Some of my suggestions are as follows;

-In introduction, a general information can be given as a result of the literature review on the subject.

-There are some grammatical and writing mistakes, please correct them. Apply to an English editing center.

- Have you used a method for sample selection? Please explain.

- When separating decimals, use dots instead of commas (in table 1).

-Include the superior aspects and differences of your study at the end of the article's discussion section.

-You can expand the discussion according to results.

-Please add a ”conclusion” subheading at the end of the “discussion”.

- There are references that are not in the journal writing format. (Check out the Author Guidelines)

Reviewer #2: This is an interesting article from a well established group that has been working for a long time to improve engagement with community pharmacies in Italy as a path for managing chronic illnesses. This is an important strategy that is relevant to many other settings around the world.

Unfortunately, this intervention study is difficult to assess since plans to assess improvements in adherence to medications were limited by low return rates for followup interviews. This doesn’t take away from the overall impressive achievement of harmonizing efforts across many community pharmacies (including with a centralized database) but it does raise several questions that I think need to be clarified:

First, in terms of framing, I assume thiswas this a pilot/feasibility study. Because if it was planned formal effectiveness study then some of the findings here (such as the proportion of non-adherent individuals, the low return rates) would have been anticipated in intervention planning through sample/power calculations, preliminary data etc). So one way to help frame the paper would be to focus more on these lessons learned for intervention planning, which will have an impact on future studies.

The evaluation scheme is biased toward adherent individuals, since it assesses individuals with a prescription in hand and who are willing to be interviewed. One way to more formally get at the overall reach of the project would be to a formal study flow diagram, so we can see the proportion of refusals /reasons for refusal at each stage. For example, if 2000 individuals were asked to participate, but only 1000 participated (as roughly shown here) then that already helps us see where the potentially nonadherent individuals may have been. A flow diagram will allow us to better visualize the project activities all the way from pharmacy enrollment to followup interview.

Study locations could be better described - there are three, but one is described as a region and two as provinces. These seem to be different sized administrative units, so a better sense of the organization would be helpful.

Description of the Morisky scale and the clinic guidelines adherence questionnaire would be helpful - perhaps as a table or appendix with these instruments. This for those who are not familiar with them. i had to google the Morisky scale in order to remember what the four items are. This should be readily available to the readers.

The GCI metric needs to be defined in the paper - I believe this is a metric the authors themselves have come up with by looking at the reference list and associated abstracts, but a stand-alone clear definition in this manuscript is needed

I do not understand the decision to exclude insulin users from the overall nonadherence analysis, since this is one-fourth of the total population. The overall nonadherence rates are both low, but they are not markedly different (8 vs 12%) and oral v. insulin therapy could be included in the model

The tables are complex, in part because continuous variables have been categorized (age, years with diabetes). I would recommend presenting this in a standard mean/median +/- SD or IQR format with statistical tests appropriate for continuous statistics. I think the categorization also leads to some likely over-interpretation of the data (U shaped curve). In addition table readability would be improved by eliminated the n under each variable (number of men/women); this is not needed as the total N is given at the top of each column. Also, for binary variables (e.g yes/no) only one category is needed, the other is implied.

In the regression models, again, it would be better to leave the continuous variables (age, years with diabetes) as continuous in the models rather than categorizing them, especially as the categories do not have any immediately obvious relevance.

I believe it would be better to report the final interview/outcome data, which is available for one-third of the population. I think the main issue here is not primarily the low return rate but rather that nonadherence was low at baseline, something which was not anticipated in planning the intervention and in considering the needed sample size. So even if all of the participants had returned for interviews, this would remain true. So at least reporting the description output from these interviews would be useful

Another thing that the authors should consider is reporting the raw scores from Morisky questionnaire. This goes to my comment above several times about over-categorizing/dichotomizing outcomes. So reporting the Morisky score (mean or median + SD or IQR) and consider changing the outcome/regression analysis to the questionnaire scores rather than a dichotomous analysis. This may give more analytical power for both the baseline and the endline analysis.

Reviewer #3: The manuscript is well described and describes how they assessed the non-adherence to drug therapy or the frequency of clinical assessment (whether or not in accordance with a guideline) of diabetic patients seen in community pharmacies. I found the paper interesting, but I felt the need for some adjustments in relation to the expectations and what was actually accomplished.

1- Objective and Conclusion

The objective states that an intervention program to monitor or enhance adherence to guidelines for pharmacological treatment will be evaluated, but what was evaluated was adherence to the prescribed medication and not to the pharmacological treatment recommended in the clinical guideline according to the patient's clinical condition. I understand that two things were evaluated: 1- adherence to the pharmacological therapy prescribed; and 2- adherence to the clinical evaluation (measured according to assessments of glycated hemoglobin and other parameters, as recommended in the guideline).

Also, this being the purpose, it was expected that the conclusion would be about the success or otherwise of the program or the results obtained from the analysis of adherence but what the authors conclude is not consistent with the objective of the study.

Regarding on conclusion:

A- the authors conclude that the community pharmacy would be an appropriate place to intercept individuals in need of health promotion intervention but it was not the purpose of the study to assess this;

B- the authors then conclude that "poor adherence to clinical guidelines is not easy to identify" and that this would be related to the low effectiveness of the intervention... but again, this aspect was not evaluated in the study, nor does it make sense since the lack of adherence was identified and described and the lack of effectiveness had more to do with the fact that the vast majority of patients did not return after 3 months for follow-up.

The abstract as well as the discussion and conclusion of the study would need to be adequate for the reader to have more clarity about what was evaluated and what conclusions he or she can draw from this study.

2- Satisfaction questionnaire

In methods, the authors mention that they applied a satisfaction questionnaire to the pharmacists without giving further details. In results, there is no description of the number of pharmacists who answered the questionnaire or any other information about it. In discussion, the authors address the subject again, but this is not a result that the reader can adequately understand given the absence of details about what was collected, in what form, and what the results of this analysis were. I suggest excluding these mentions or describing them appropriately in all sections of the manuscript.

6. PLOS authors have the option to publish the peer review history of their article (what does this mean?). If published, this will include your full peer review and any attached files.

Reviewer #1: No

Reviewer #2: **Yes: **Peter Rohloff

Reviewer #3: No

---

## [Author Response · Author response to Decision Letter 0]

5 Jul 2021

Dear Editor and Reviewers,

we are pleased to submit our revised manuscript for your consideration for publication. 

We thank you and the reviewers for the careful consideration that was given to the original version of the manuscript. We have addressed the issues raised by each reviewer.

Below in the text each issue has been discussed. The revised text has been approved by the co-authors.

Finally, the manuscript has been revised by a mother tongue speaker. 

Please let us know if you have any additional questions. Thank you for the opportunity to revise this manuscript. 

Best regards,

Francesca Baratta

Journal Requirements:

We thank you. We checked the manuscript and we modified it when needed.

2. We note that you used the Morisky Medication Adherence Scale (MMAS-8) in your study. It is our understanding that this scale is protected by copyright and requires a license agreement for use. Please explain in your Methods section whether you obtained permission and a license agreement for the use of the MMAS-8 in your study. In addition, we note that the scale is reproduced as Supporting Information. Due to copyright restrictions, please remove this from your submission.

You are right; the MMAS-8 is in fact protected by copyright. However, we used a 4-item scale derived from the Italian version of the 8-item questionnaire, which was included in a wider questionnaire about use of drugs. We have clarified it in the manuscript and we deleted references to Morisky in the methods in order to avoid misunderstanding. Indeed, one of our conclusions was that the use of the original 8-item Morisky scale could be more helpful for future studies. Hence, also to respond to a reviewer’s request who asked to see the questionnaires, we left the complete forms in the appendix (lines 130-137).

Thank you for these indications and for updating the statement. We have uploaded an Excel file with the minimal anonymized data set as Supporting Information.

We moved the ethics statement in the Methods section as requested and corrected an imprecision (lines 178-186).

We modified as suggested (lines 144, 157, 193, 425-429).

 

Reviewers' comments:

Reviewer's Responses to Questions

Comments to the Author

1. Is the manuscript technically sound, and do the data support the conclusions?

Reviewer #1: Yes

Reviewer #2: Partly

Reviewer #3: Yes

2. Has the statistical analysis been performed appropriately and rigorously?

Reviewer #1: Yes

Reviewer #2: No

Reviewer #3: Yes

3. Have the authors made all data underlying the findings in their manuscript fully available?

Reviewer #1: Yes

Reviewer #2: No

Reviewer #3: Yes

4. Is the manuscript presented in an intelligible fashion and written in standard English?

Reviewer #1: Yes

Reviewer #2: No

Reviewer #3: Yes

5. Review Comments to the Author

Reviewer #1:

Dear Authors,

Manuscript, named as “Monitoring adherence to clinical guidelines among patients with type 2 diabetes in community pharmacies. Results from an experience in Italy” was sended me to review.

Study is very valuable in terms of study design and contribution to the literature.

We thank you for your appreciation.

Some of my suggestions are as follows;

-In introduction, a general information can be given as a result of the literature review on the subject.

Thanks for the suggestion. We added some sentences in the Introduction. More details have been left in the discussion (lines 73-75, 80-82).

-There are some grammatical and writing mistakes, please correct them. Apply to an English editing center.

We had English proofread by a native speaker.

- Have you used a method for sample selection? Please explain.

We are sorry it wasn’t clear: all patients entering the pharmacy were invited. Given the low expected number of daily accesses of diabetic patients, no sampling was applied. We have added this specification in the text (line 118). 

- When separating decimals, use dots instead of commas (in table 1).

Thank you for noticing it, we corrected the tables.

-Include the superior aspects and differences of your study at the end of the article's discussion section.

We thank you for the suggestion. In accordance with what you suggest below, and also to accomplish other reviewers’ requests, at the end of the discussion we have added a specific paragraph of conclusions, highlighting the main lessons learned from this study, which include what we consider the novelties and the implications of our study (lines 358-373). 

-You can expand the discussion according to results.

We have expanded the summary of our results at the beginning of the discussion, so that the subsequent paragraphs can now be more clearly linked to our results. We have also added subheadings, to help follow the reasoning. We hope that this reorganization of the discussion will satisfy the points you raised (lines 261-350). 

-Please add a ”conclusion” subheading at the end of the “discussion”.

As mentioned before, we have added the suggested subheading (line 354).

- There are references that are not in the journal writing format. (Check out the Author Guidelines)

We thank you for the remark. We revised the references and we corrected them when needed. 

Reviewer #2: 

This is an interesting article from a well established group that has been working for a long time to improve engagement with community pharmacies in Italy as a path for managing chronic illnesses. This is an important strategy that is relevant to many other settings around the world.

We thank you for appreciating our work.

Unfortunately, this intervention study is difficult to assess since plans to assess improvements in adherence to medications were limited by low return rates for follow-up interviews. This doesn’t take away from the overall impressive achievement of harmonizing efforts across many community pharmacies (including with a centralized database) but it does raise several questions that I think need to be clarified:

First, in terms of framing, I assume this was this a pilot/feasibility study. Because if it was planned formal effectiveness study then some of the findings here (such as the proportion of non-adherent individuals, the low return rates) would have been anticipated in intervention planning through sample/power calculations, preliminary data etc). So one way to help frame the paper would be to focus more on these lessons learned for intervention planning, which will have an impact on future studies.

Thanks for these comments and suggestions. As you correctly remark, we aimed to assess the transferability of the programme. We have indeed rephrased the conclusions in terms of lessons learned, which we hope can be of greater help to readers (lines 358-373).

The evaluation scheme is biased toward adherent individuals, since it assesses individuals with a prescription in hand and who are willing to be interviewed. One way to more formally get at the overall reach of the project would be to a formal study flow diagram, so we can see the proportion of refusals /reasons for refusal at each stage. For example, if 2000 individuals were asked to participate, but only 1000 participated (as roughly shown here) then that already helps us see where the potentially nonadherent individuals may have been. A flow diagram will allow us to better visualize the project activities all the way from pharmacy enrollment to followup interview.

Since there were only two reasons for exclusion (being a type 1 diabetes patient and refusal to give consent), we added these two numbers and proportions in the text. We included the proper flow diagram as Supplementary Information (lines 192-193).

Study locations could be better described - there are three, but one is described as a region and two as provinces. These seem to be different sized administrative units, so a better sense of the organization would be helpful.

We thank you for the comment. We used different terms (region, province, health unit, district) only because of Italian administrative reasons, but this did not interfere with our project. It is true that at an international level this classification may be unclear, therefore we simplified the manuscript. (lines 104-106).

Description of the Morisky scale and the clinic guidelines adherence questionnaire would be helpful - perhaps as a table or appendix with these instruments. This for those who are not familiar with them. i had to google the Morisky scale in order to remember what the four items are. This should be readily available to the readers.

We understand that we were imprecise and clarified that we used a 4-item scale, modified from the Italian version of the original 8-item Morisky scale (lines 130-134). The version we used was provided in the appendix, together with the other questionnaires. 

The GCI metric needs to be defined in the paper - I believe this is a metric the authors themselves have come up with by looking at the reference list and associated abstracts, but a stand-alone clear definition in this manuscript is needed.

Yes, you are right, the GCI metric has been in fact developed by one of the authors, together with other Italian diabetologists, and it has been used for many years. Its definition is at lines 166-169.

I do not understand the decision to exclude insulin users from the overall nonadherence analysis, since this is one-fourth of the total population. The overall nonadherence rates are both low, but they are not markedly different (8 vs 12%) and oral v. insulin therapy could be included in the model.

Thank you for this comment, we have now clarified that the decision was taken mainly because the distribution of non-adherence by individual characteristics among insulin users was reversed compared to oral drug users, and therefore a pooled analysis was impossible (lines 203-206).

The tables are complex, in part because continuous variables have been categorized (age, years with diabetes). I would recommend presenting this in a standard mean/median +/- SD or IQR format with statistical tests appropriate for continuous statistics. I think the categorization also leads to some likely over-interpretation of the data (U shaped curve). In addition table readability would be improved by eliminated the n under each variable (number of men/women); this is not needed as the total N is given at the top of each column. Also, for binary variables (e.g yes/no) only one category is needed, the other is implied.

In the regression models, again, it would be better to leave the continuous variables (age, years with diabetes) as continuous in the models rather than categorizing them, especially as the categories do not have any immediately obvious relevance.

The issues raised in these two paragraphs are interrelated and we will try to address them jointly.

Following your concern about the categorization of continuous variables, we tested the linearity of age and years of diabetes with respect to all the outcome measures of non-adherence. We found that deviation from linearity was always statistically significant, except for age vs. clinical examinations (where the age estimate was always non-significant even when it was left linear). Therefore, we believe that we cannot perform linear models with the continuous variables and we have kept them categorized. Therefore, we also left in tables 1 and 2 the same categorization used in the models, in order to show raw data for each category.

Indeed, we are aware that the tables can be complex to read, however we could not modify them as suggested, because the percentages in table 2 are not frequency distributions, but prevalence of non-adherence in each category and they cannot be derived from other numbers, because the denominators (e.g. the number of women insulin users) are not reported elsewhere.

I believe it would be better to report the final interview/outcome data, which is available for one-third of the population. I think the main issue here is not primarily the low return rate but rather that nonadherence was low at baseline, something which was not anticipated in planning the intervention and in considering the needed sample size. So even if all of the participants had returned for interviews, this would remain true. So at least reporting the description output from these interviews would be useful.

Thank you for the suggestion. We added the results of adherence at follow-up at lines 245-249.

Another thing that the authors should consider is reporting the raw scores from Morisky questionnaire. This goes to my comment above several times about over-categorizing/dichotomizing outcomes. So reporting the Morisky score (mean or median + SD or IQR) and consider changing the outcome/regression analysis to the questionnaire scores rather than a dichotomous analysis. This may give more analytical power for both the baseline and the endline analysis.

For the 4-item scale the same linearity problem arises, which does not allow us to analyse the score as continuous. In fact, the high adherence rates shown in the article are reflected in a skewed distribution, with nearly all patients answering 0 or 1 and very few (around 3%) with a score >2.

However, we understand your point and therefore we performed a sensitivity analysis using a different categorization of the scale, which classified patients with a score >0 as non-adherent (as opposed to a score >1). The model results showed substantially the same results as the previous ones. We have reported the result of this sensitivity analysis at lines 346-348.

 

Reviewer #3: The manuscript is well described and describes how they assessed the non-adherence to drug therapy or the frequency of clinical assessment (whether or not in accordance with a guideline) of diabetic patients seen in community pharmacies. I found the paper interesting, but I felt the need for some adjustments in relation to the expectations and what was actually accomplished.

We thank you for your appreciation.

1- Objective and Conclusion

The objective states that an intervention program to monitor or enhance adherence to guidelines for pharmacological treatment will be evaluated, but what was evaluated was adherence to the prescribed medication and not to the pharmacological treatment recommended in the clinical guideline according to the patient's clinical condition. I understand that two things were evaluated: 1- adherence to the pharmacological therapy prescribed; and 2- adherence to the clinical evaluation (measured according to assessments of glycated hemoglobin and other parameters, as recommended in the guideline).

Also, this being the purpose, it was expected that the conclusion would be about the success or otherwise of the program or the results obtained from the analysis of adherence but what the authors conclude is not consistent with the objective of the study.

Regarding on conclusion:

A- the authors conclude that the community pharmacy would be an appropriate place to intercept individuals in need of health promotion intervention but it was not the purpose of the study to assess this;

B- the authors then conclude that "poor adherence to clinical guidelines is not easy to identify" and that this would be related to the low effectiveness of the intervention... but again, this aspect was not evaluated in the study, nor does it make sense since the lack of adherence was identified and described and the lack of effectiveness had more to do with the fact that the vast majority of patients did not return after 3 months for follow-up.

The abstract as well as the discussion and conclusion of the study would need to be adequate for the reader to have more clarity about what was evaluated and what conclusions he or she can draw from this study.

We thank you for the valuable comments and agree with you. We modified the text, in particular the Title, the Abstract and the Introduction, to clarify the objectives. We then changed the Conclusions, to make them more coherent with the objectives.

2- Satisfaction questionnaire

In methods, the authors mention that they applied a satisfaction questionnaire to the pharmacists without giving further details. In results, there is no description of the number of pharmacists who answered the questionnaire or any other information about it. In discussion, the authors address the subject again, but this is not a result that the reader can adequately understand given the absence of details about what was collected, in what form, and what the results of this analysis were. I suggest excluding these mentions or describing them appropriately in all sections of the manuscript.

We appreciate your suggestion. We consider this qualitative evaluation made by pharmacists an added value of our project. Therefore, we have expanded the paragraph in the Materials and Methods section to better explain the satisfaction questionnaire (lines 151-157), and included the questionnaire in the Supporting Information files. Furthermore, we gave the requested number of respondents in the Results (line 251) and commented the findings both in the discussion and in the conclusions (lines 326-328, 370-373).

6. PLOS authors have the option to publish the peer review history of their article (what does this mean?). If published, this will include your full peer review and any attached files.

Do you want your identity to be public for this peer review? For information about this choice, including consent withdrawal, please see our Privacy Policy.

Reviewer #1: No

Reviewer #2: Yes: Peter Rohloff

Reviewer #3: No

---

## [Decision Letter · Decision Letter 1]

9 Aug 2021

Monitoring adherence to pharmacological therapy and follow-up examinations among patients with type 2 diabetes in community pharmacies. Results from an experience in Italy

PONE-D-21-09684R1

Dear Dr. Baratta,

We’re pleased to inform you that your manuscript has been judged scientifically suitable for publication and will be formally accepted for publication once it meets all outstanding technical requirements.

Kind regards,

Filipe Prazeres, MD, MSc, Ph.D.

Academic Editor

PLOS ONE

Additional Editor Comments (optional):

Reviewers' comments:

Reviewer's Responses to Questions

**Comments to the Author**

1. If the authors have adequately addressed your comments raised in a previous round of review and you feel that this manuscript is now acceptable for publication, you may indicate that here to bypass the “Comments to the Author” section, enter your conflict of interest statement in the “Confidential to Editor” section, and submit your "Accept" recommendation.

Reviewer #2: All comments have been addressed

Reviewer #3: All comments have been addressed

2. Is the manuscript technically sound, and do the data support the conclusions?

Reviewer #2: (No Response)

Reviewer #3: Yes

3. Has the statistical analysis been performed appropriately and rigorously? 

Reviewer #2: (No Response)

Reviewer #3: Yes

4. Have the authors made all data underlying the findings in their manuscript fully available?

Reviewer #2: (No Response)

Reviewer #3: Yes

5. Is the manuscript presented in an intelligible fashion and written in standard English?

Reviewer #2: (No Response)

Reviewer #3: Yes

6. Review Comments to the Author

Reviewer #2: (No Response)

Reviewer #3: The changes made the manuscript look much better. I consider the paper to be suitable for publication.

7. PLOS authors have the option to publish the peer review history of their article (what does this mean?). If published, this will include your full peer review and any attached files.

Reviewer #2: **Yes: **Peter Rohloff

Reviewer #3: No